

# Brief communication: CMIP6 does not suggest any circulation change over Greenland in summer by 2100

Alison Delhasse [1], Edward Hanna [2], Christoph Kittel [1], and Xavier Fettweis [1]

[1]Laboratory of Climatology, Department of Geography, SPHERES, University of Liège, Liège, Belgium
[2]School of Geography and Lincoln Centre for Water and Planetary Health, University of Lincoln, Lincoln, UK

*Correspondence to:* Alison Delhasse (alison.delhasse@uliege.be)

**Abstract.** The Greenland blocking index (GBI), an indicator of the synoptic-scale circulation over Greenland, has been anoma-lously positive during summers since the late 1990s. Such changes in atmospheric circulation have led to an increase in Green-land summer temperatures, a decrease in cloud cover and greater surface melt. The GBI is therefore a key indicator of melting and surface mass balance variability over the Greenland ice sheet. However, the fifth phase of the Coupled Model Intercom-
parison Project (CMIP5) models do not represent any increase in GBI as suggested by observations. Until 2100, no significant long-term trend in the GBI, and therefore no circulation changes, are projected. In this study the new generation of CMIP6 Earth-system models is evaluated in order to analyze the evolution of the future GBI. All CMIP5 and CMIP6 projections reveal the same trend towards a decrease of the GBI until 2100 and no model reproduces the strong increase in GBI observed over the last few decades. Significant melting events related to a highly positive GBI, as observed this summer 2019, are still not
considered by CMIP6 models and therefore the projected surface melt increase of the ice sheet is likely to be underestimated if such circulation changes persist in the next decades.

## 1 Introduction

An extreme melting event occurred this summer 2019 over Greenland and followed several other exceptional years ((eg., 2003, 2005, 2007, 2008, 2010 and 2012, Fettweis et al., 2013a; Tedesco et al., 2013; Hanna et al., 2014)). This 2019 melt season
resulted from specific anticyclonic conditions, particularly advection of warm and moist air along the western part of the ice sheet towards the north and enhanced the absorption of solar radiation which strengthens melt, itself amplified by the melt-albedo feedback (MAF) (Tedesco and Fettweis, 2019). Although this near-record of melt is also due to the combination of anticyclonic conditions with low winter accumulation, it is part of a series of Arctic summers characterized by blocking events which have become increasingly frequent over the past 20 years (Fettweis et al., 2013a; Hanna et al., 2014; Belleflamme et al.,
2015). These more frequent blocking events reveal a change in the mean atmospheric summer circulation in Greenland that is one of the main contributors to the recently observed acceleration in surface melt (Fettweis et al., 2013b; Overland et al., 2012). This has contributed to the decrease in the surface mass balance (SMB) since the 1990s (van den Broeke et al., 2016; Fettweis et al., 2017; Trusel et al., 2018; Noël et al., 2019).





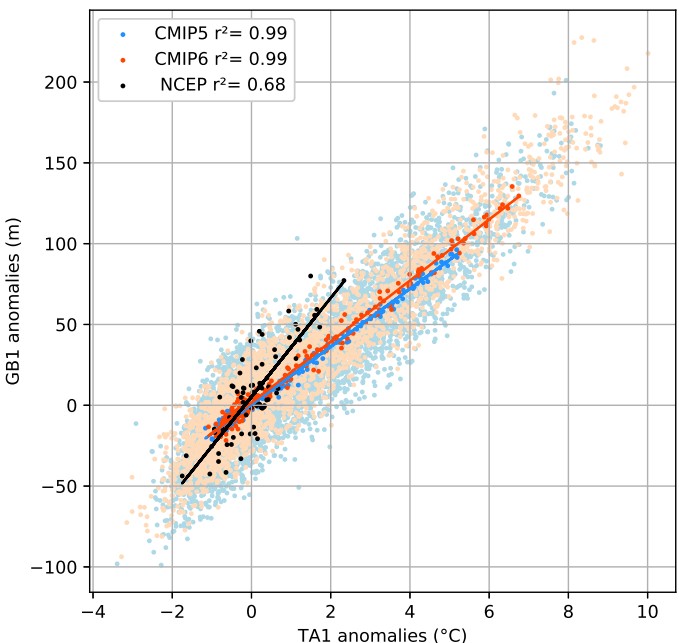

**Figure 1.** Relation between Greenland Blocking Index (GB1) anomalies and mean 850, 700 and 500 hPa temperature anomalies over 60–80 ° N, 20–80 ° W. The reference period is 1986-2005.

Despite the importance of such circulation changes, they are not represented by any of the CMIP5 Earth-system models (ESM) (Fettweis et al., 2013b; Hanna et al., 2018a). None of the models suggest any increase in Greenland blocking events by 2100, while some models even suggest moderate decreases in blocking. This model-observation disparity means that the melt increase over the Greenland ice sheet due to global warming could be underestimated by a factor of two, suggesting a potential 5 underestimation of the future SMB decrease (Delhasse et al., 2018).

The latest Climate Model Intercomparison Project (CMIP6) provides new projections simulated by an improved generation of ESMs. The most important enhancements compared to CMIP5 are a higher resolution, a more sophisticated physics notably due the improvement of the coupling between the different components of the Earth-system, and better constrained concentrations of aerosols and other near-term climate forcings (Eyring et al., 2016; O'Neill et al., 2016; Voldoire et al., 2019). While the 10 CMIP6 models are forced by similar atmospheric greenhouse gas scenarios, they are more sensitive than CMIP5 as they show a stronger warming at the end of the Twenty First century. This could partly result from stronger cloud feedbacks (Andrews et al., 2019; Gettelman et al., 2019; Voldoire et al., 2019). For the Greenland ice sheet, a new investigation suggests a doubling of surface melt and a lengthening of the melt season in CMIP6 models relative to CMIP5 (Hofer et al. 2019 submitted).





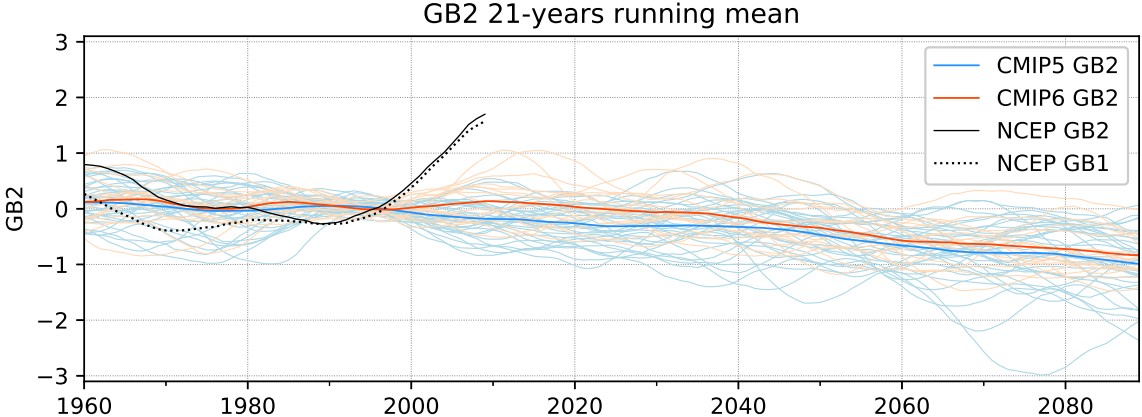

**Figure 2.** JJA GB1 (dashed black line) and GB2 (solid black line, defined in Eq. 1) indices over 1950–2100 as simulated by NCEP/NCAR Reanalysis 1 as well as by all the CMIP5 models (RCP8.5 scenario, blue lines) and the CMIP6 models (ssp585, red lines).The historical scenario is used over 1950 until 2005 for CMIP5 and 2014 for CMIP6, while RCP8.5 and ssp585 respectively afterwards. A 21-year running mean has been applied to smooth the time series, and values have been normalized using 1986–2005 as the reference period.

Considering the impact of such blocking events and the availability of new CMIP6 ESM projections, the aim here is to assess: 1) the ability of these simulations to represent the recent increase in Greenland blocking, and 2) whether such circulation/blocking changes are predicted from now until 2100.

## 2 Data and methodology

5 The summer (JJA) Greenland Blocking Index (GBI) is used to assess the representation in CMIP6 and CMIP5 models of the recent summer blocking events observed over Greenland. GBI is defined as the area-weighted mean geopotantial height of the 500-hPa level (Z500) over the Greenland area 60–80 ° N, 20–80 ° W (GR) and is referred to as GB1 hereinafter (Hanna et al., 2016). In order to avoid the influence of the global temperature increase (Figure 1) and study only the dynamic (and not thermal) atmospheric changes that occur above Greenland, the GB2 index (Eq. 1, Hanna et al., 2018a) is used here. GB2 is 10 calculated as the normalized difference between GBI and the area-weighted mean Z500 over the northern hemisphere region 60-80 °N (NH).

$$GB1 = Z500_{GR} \qquad\qquad GB2 = Z500_{GR} - Z500_{NH} \qquad\qquad (1)$$

The free-atmosphere temperature related to the GBI region (TA1 hereafter, Eq. 2) is the representation in a model of one of the factors driving and influenced by Z500 variability. Similarly to GBI, we used here TA2 which is defined as the difference of



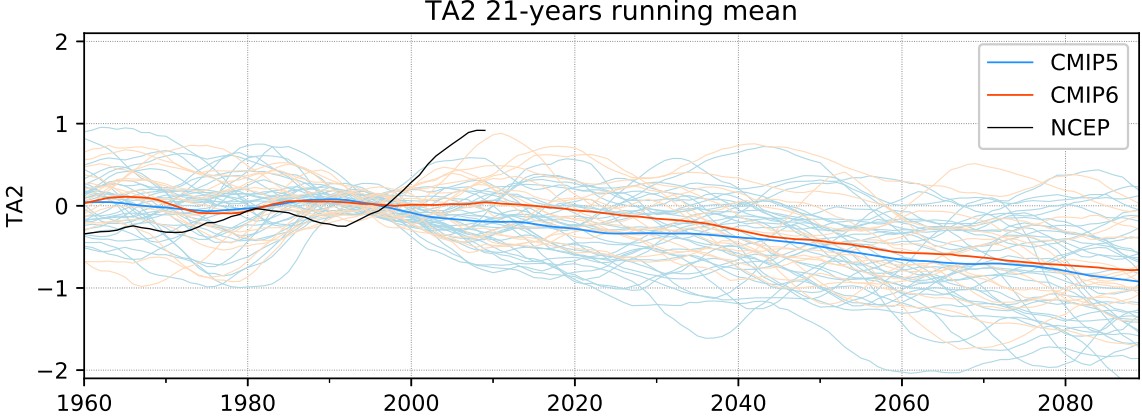

**Figure 3.** Similar to Fig. 2 but showing TA2 (defined in Eq. 2). Values are normalized considering 1986 -– 2005 as reference period.

monthly mean temperature at 850, 700 and 500 hPa between the GR and NH areas (Eq. 2). TA2 is chosen for the same reason than GB2 and is used in its normalized form.

$$TA1 = \frac{(T850 + T700 + T500)_{GR}}{3} \qquad\qquad TA2 = TA1 - \frac{(T850 + T700 + T500)_{NH}}{3} \qquad (2)$$

TA2 and GB2 are based on summer CMIP5 and CMIP6 simulations (1950 – 2100) and are compared with the same parame-
ters based on NCEP/NCARv1 reanalysis (Kalnay et al., 1996, NCEP hereinafter) for the following recent period: 1950 – 2019.
Only CMIP6 models with outputs available for the ssp585 scenario (Gidden et al., 2019) are considered in this study. The
CMIP5 models used in Hanna et al. (2018a) (considering the RCP8.5 scenario only) are also included. The new CMIP6 sce-
nario, ssp585, is approximately equivalent to RCP85 in term of radiative forcing (+ 8.5 W/m$^2$, Taylor et al., 2012; Eyring
et al., 2016; O'Neill et al., 2016). For comparison, the NCEP reanalysis is used as a reference for recent observed climate
(1950 – 2019).

All time series are presented here in normalized form with respect to the reference period 1986 – 2005 as in Hanna et al.
(2018a). A weighted running mean is applied by using a midpoint-centered 21-year running mean, which shortens the begin-
ning and the end of series by 10 years, as shown in our figures.

## 3  Results

Figure 1 plotting TA1 vs GBI with respect to the reference period, shows that the GBI variability in the ESMs (both CMIP5 and
CMIP6) is closely linked to atmospheric (mid-troposphere) temperature changes. However, the observed GBI increases more
strongly despite surface/lower atmosphere warming, due to a shift towards more favourable atmospheric dynamic (jet stream)
precursors causing Greenland blocking events. Figure 1 also shows that the main difference between CMIP5 and CMIP6 is the
stronger CMIP6 warming rate.





The NCEP reanalysis shows a significant increase of GB2 (larger than the interannual variability) for the recent period, which is not represented by any of the CMIP5 and CMIP6 models (Figure 2). In CMIP5, the highest GB2 value does not even reach 1 standard deviation (std). Until 2020 two CMIP6 models (MRI ESM2-0 and EC-Earth3) reach 1 std of GB2 and oscillate until the end of the century with a decreasing trend. Until 2040, two other CMIP6 models (EC-Earth3-Veg and NESM3) oscillate
near 1 std. Nevertheless, at the end of the century, all the ESMs project a decrease in GB2 and indicate no permanent circulation changes or blocking increase, unlike what has recently occurred.

Similarly to GB2, the recent observed increase in TA2 (based on reanalysis) is mainly not represented by the CMIP6 ESMs (Figure 3). Only one CMIP6 (MRI-ESM2-0) model suggests a temporary increase in TA2 during ten years as large as that of the NCEP record. The largest GB2 oscillations mentioned above are mainly due to the temperature variability and not caused
by persistent circulation changes. This also suggests that the recent circulation anomalies are not due to natural variability. The same models show as intense TA2 oscillations as GB2 whereas NCEP GB2 increase is more intense than TA2 which is not significant (< 1 std).

Table S1 (in supplementary materials) summarizes occurrences of summers characterized by GB2 > (<) 1 (-1) and 2 (-2) for NCEP over the current period, CMIP5 and CMIP6 models (2000-2019, 2040-2060 and 2080-2100). Between 2000 and 2019,
the NCEP summer GB2 has been negative six times but only two of these instances occurred since 2007. Out of 2007 to 2019 summers, only 2013, 2017 and 2018 had GB2 < 1 (-2.37, 0.22 and -2.07 respectively, Figure S1 in supplementary materials). For CMIP5 models, the maximum number of occurrences of GB2 > 1 during 2000 – 2019 is six for the MPI-ESM-MR model, and only few models (eleven) have summers with GB2 > 2. On the other hand, CMIP6 models seem to show higher summer occurrences of GB2 > 1 over 2000 – 2019 relative to CMIP5; however, the maximum occurrence (nine with NESM3), does not
reach that of NCEP (eleven). Up to the end of the century there is a decrease in the occurrence of positive GB2, which supports the previous findings, and there is no significant difference between CMIP5 and CMIP6.

Note that the results presented in this study are insensitive to changing both the chosen reference period (see in supplementary material Fig. S5 and S6) and the length of the running mean (see in supplementary materials Fig. S2 to S4).

## 4 Conclusions

Blocking events have often characterized summers in Greenland since the late 1990s, leading to several melting records with the latest one in 2019. This reflects particular atmospheric circulation conditions as gauged by the GBI index. The GBI has strongly increased since the late 1990s but yet the previous generation of ESM models (CMIP5) do not show such an increase (Hanna et al., 2018a). In this study we have used GB2 to evaluate the ability of the new CMIP6 models to represent recent variability of the atmospheric circulation over Greenland in summer, as well as its future evolution to 2100. We conclude that
no circulation changes are represented for recent or future climates (1990s to 2100) by the CMIP6 models, and that the free-atmosphere temperature variability fully drives the GB2 changes in the ESM-based projections. This suggests that, according to CMIP5 and CMIP6 models, recent circulation anomalies are not due to natural variability.





As shown in Delhasse et al. (2018), atmospheric warming combined with recently observed circulation changes could result in a doubling of the surface melt increase of the Greenland ice sheet. By not resolving a possible sustained increase in future Greenland blocking events and associated atmospheric circulation changes, CMIP6 ESM-based projections have a distinct potential to underestimate Greenland ice sheet surface melt if the observed recent circulation anomalies persist in the next

decades. Major uncertainties concerning the global understanding of the processes producing these blocking events do not enable reliable simulation of their occurrence by climate models (Woollings et al., 2018). Therefore, future ESM developments should focus on representing these changes in atmospheric circulation in order to reduce the uncertainty in: (1) projections of Greenland ice sheet contribution to global sea level rise; and (2) possible downstream effects of Greenland blocking on the North Atlantic polar jet stream and accompanying extreme weather conditions over Northwest Europe (Hanna et al., 2018b).

*Author contributions.* AD, EH and XF designed the study. AD did the analysis and wrote the manuscript while CK, XF and EH provided advice and discussed concepts. All authors discussed and revised the final version of the manuscript.

*Competing interests.* The authors declare that they have no conflict of interest.

*Acknowledgements.* We acknowledge the World Climate Research Programme's Working Group on Coupled Modelling, which is responsible for CMIP, and we thank the climate modelling groups for producing and making available their model output. For CMIP the U.S.

Department of Energy's Program for Climate Model Diagnosis and Intercomparison provides coordinating support and led development of software infrastructure in partnership with the Global Organization for Earth SystemScience Portals. NCEP-NCAR-v1 reanalysis are provided by the NOAA/OAR/ESRL PSD, Boulder, Colorado, USA, from their Web site at https://www.esrl.noaa.gov/psd/.



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
