# Peer review of "Brief communication: CMIP6 does not suggest any circulation change over Greenland in summer by 2100"

_The Cryosphere, 2019_

## Referee Comment (RC1) · Anonymous Referee #1 · 2 Mar 2020

major comments Figure 1 illustrates an important result featured in this study.

Do any individual models capture more realistic Greenland blocking temporal patterns, i.e. persistence in extreme Greenland blocking? I see page 4 line 19 "NESM3" is referred to in this regard.

I am of the position that using GCM ensembles is a poor strategy because of destructive interference among models. Rather, one should identify which models among the ensemble simulate the phenomenon in question and then gain insight by focusing on why the more skilfull models contrast with the others. To average all models then drive a message that no models capture observations is therefore disingenuous and misled,

especially when individual models e.g. NESM3 referred to in the study may capture the phenomenon in question?

For robust climatological inferences, 30 year periods are expected. Yet, here presented are 21 year (running) periods. 21 years need be justified.

Why should we expect Greenland blocking index (GBI) to continue trending positive?

Problematic statement. "The Greenland blocking index (GBI), an indicator of the synoptic-scale circulation over Greenland, has been anomalously positive during summers since the late 1990s." Fig 2 does not support that statement. Then there are annual exceptions: 2013, 2018, 2017

How does the study reaches the conclusion "free atmosphere temperature variability fully drives the GB2 changes"?

minor comments io == instead of

Figs 2, 3 "21-year" io "21-years"

page line number

1 9 delete "this" 1 22 define "This" 2 6 "advanced" io "improved" 2 7 "advancements" io "enhancements" 3 1 "Greenland" io "such" 3 3 "CMPIP6" io "these" 3 2 delete "such" 3 3 "until year 2100." 5 10 define "This" 5 18 "more frequent" io "higher" 5 21 "no significant difference between CMIP5 and CMIP6 ensembles." 5 25 rather than the vague "often", state a frequency of melt cases above say 1 standard deviation of a 30 year period 5 10 define "This" 5 31 define "This" 5 30 "by the CMIP6 model ensemble," 6 2 "doubling of the surface melt increase of the Greenland ice sheet." by 2100? 6 3 delete "distinct" 6 5 define "these" 6 6 "global climate models" 6 7 define "these changes"

---

## Referee Comment (RC2) · Anonymous Referee #2 · 27 Mar 2020

This manuscript provides an update to analysis of CMIP5 data which will probably be of use. I'm sorry not to be able to provide a more positive review, but there are many ways that this could be improved. I'm not familiar with the journal though, so perhaps for a brief communication this is acceptable. However, several of the stated conclusions are unsupported by the analysis, so at present I would say that the manuscript is not suitable for publication with these in.

Unsupported statements:

- 'any circulation change' in the title. This clearly hasn't been demonstrated, as only one very simple circulation measure is used in the whole paper.

[Figure]

- 'Significant melting events related to a highly positive GBI...' in abstract. The paper focuses on 21-year running means, so seasonal extremes have not been considered.

- 'a shift towards more favourable atmospheric dynamic precursors...' on p4. No analysis of this is given.

- lines 9,10 on p5: no separation between 'temperature variability' and 'circulation changes' has been attempted, and statements like this are very vague.

- line 30-32, p5: No analysis has been done on what 'drives' the GB2 'changes'.

Major points:

While simplicity is a good thing in general, some of the approaches here could be too simple. The 'blocking' indices are really just seasonal mean height diagnostics, so can't be said to reflect blocking necessarily. There are probably many other things mixed in to these indices, such as hemispherically symmetric annular mode variability. It's quite possible that the interannual variability of such a time series does represent blocking, but that trends could be dominated by a different process, perhaps a thermodynamic one. I would be tempted to remove the word 'blocking' from the paper. Similarly, the temperature indices are quite arbitrary - should be mass-weighted at least.

The NCEP1 reanalysis is very old now and should no longer be used on its own. Other obvious ones to add would be ERA-20C, JRA55.

The text in many places is not very scientific. The most striking example is the frequent use of the word 'changes'. Compared to what?

The paper is also not very objective. The authors clearly already have a view on this, that summer blocking is increasing over Greenland due to anthropogenic forcing. But this is certainly not shown in the evidence they have presented. A more objective approach would allow for the possibility that multidecadal variability is underestimated in the models.

[Figure]

There is also a potential field significance issue here, in that the Greenland region has been pre-selected as a region where interesting 'trends' in height are apparent in the observations. If a 95% significance level is used, for example, 5% of Earth's surface might be expected to show a 'significant' trend in any analysis. Are the authors sure that this is not happening here?

Other questions:

Fig 1 is very interesting. Why are the GB1 and TA1 indices used for this, and how does it look for GB2 and TB2?

Fig 2 and the discussion could be clearer on the normalisation - this was done before the time filtering?

Why do the model projections show a decrease in TA2 over the 21st century. This seems very surprising given our expectations of polar amplification.

- Intro: why does increased inflow of moist air lead to increased absorption of solar radiation - this seems counter-intuitive.

- p2, line 12: Why a doubling here? The climate sensitivity is higher, but not by a factor of two...?

- 'significant increase of GB2' on p5 needs to be more rigourous. eg period?

- p6, line 2: doubling compared to what?

---

## Editor Comment (EC1) · Alison Delhasse et al. · 3 Apr 2020

Dear authors,

Two referee reports are now available, and I would like to thank the two colleagues for their time and comments.

In my access review I noted some potential issues, which I would like to express in this discussion as well since they align with referee comments. In particular, there are two things that were unclear to me.

a) My main concern is that the GB2/TA2 indices are based on the entire NH conditions - why is that? I think it is hard to understand for the reader why such a large spatial reference is ingested into these indices, and how processes elsewhere (i.e. not near Greenland) could contribute to producing the difference between reanalysis and GCMs. I regard this as a potential weakness for the conclusions. - - Referee #2 made comments in a similar direction.

b) Why is only the oldest generation of reanalysis (NCEP/NCAR) used in the study? I appreciate that it provides the longest record, but you also refer to the reference period 1986-2005 or 2000-2019 in the paper, which are both covered by the latest generation of reanalysis (Merra-2, ERA5). You must include further analysis to demonstrate that your "observations" are robust across available products. - - Again, Referee #2 argued similarly.

Moreover, I agree with Referee #1 that using a GCM ensemble is problematic for concluding on processes. It is well known that GCMs have very different skills for different regions, and in a process-oriented study those GCMs with low skill for a certain region should not be allowed to participate. I also agree with Referee #2 that simplicity is a good approach sometimes, but here the analyses are overly simplistic, especially in face of the (very!) strong conclusion expressed in the title of the paper.

In summary, both referees and myself have major concerns that the conclusions drawn in the manuscript are supported by the presented material. My decision, therefore, is to not consider this manuscript for publication in TC further.

I would like to say again that the topic is very interesting and that I hope the authors will be able to follow this in future studies, in particular by adding process-based analyses of blocking over Greenland to the "index approach".

---

## Author Comment (AC1)

Dear editor, dear reviewers,

We first would like to thank you for your comments and for taking the time to discuss our manuscript.

We are conscious that the paper in its present state needs to be improved, but we would like to address a few points that have been highlighted and that we think should not be a reason to reject the paper.

First, the paper was written as a continuation of Hanna et al (2018). It was therefore important for us to keep the same methodology and structure as in 2018. This explains the different choices that have been made here and criticized for some of them: indices studied, reference period, rolling average, etc. We would like to point out that most of them (ie., indices, reference period, rolling average) have already been discussed and answered here (https://www.the-cryosphere-discuss.net/tc-2018-91/tc-2018-91-AC2-supplement.pdf )

Secondly, we understand that the choice of a single reanalysis may be subject to questioning. If we have focused on only one it is because the choice of the reanalysis used for the comparison does not influence our conclusions, as was the case in Hanna et al. (2018) and as shown their fig 1 (and presented hereafter). We acknowledge that we should have presented in the paper that using other reanalyses does not change the results. Adding reanalyses for comparison is therefore part of the easy corrections to the analysis and does not seem to compromise the interest of the study, neither the result accuracy.

[Figure]

**Figure 1.** Time series of JJA GB1 (dashed red line) and GB2 (solid red line) indices over 1950–2100 as simulated by NCEP/NCAR Reanalysis 1 (red line), by 20CRv2c reanalysis (green line), and by ERA-20C reanalysis in blue as well as by all the CMIP5 models (grey lines) for which both RCP4.5 and RCP8.5 scenarios are available. For the CMIP5-based time series, the historical scenario is used over 1900–2005 and both RCP4.5 and RCP8.5 afterwards. A 20-year running mean has been applied to smooth the time series, and values have been normalised (average  = 0 and standard deviation = 1) using 1986–2005 as the reference period.

Then we agree that it is not the best strategy to focus only on the ESM mean rather than on their individual trends. Indeed, if one or two models have negative trends for the wrong reasons, they pull the trend downwards. We therefore analzyed each model individually and since they mostly do not differ from the mean, we decide to present and mainly discuss the average value of CMIP5 and CMIP6 models. However we highlighted the few models that have normalized GB2 values close to or above 1 (similar to the current observed normalized GB2 values). And when we say that no circulation change is represented in the future, we are not only considering the mean trend, but the fact that no ESM (when considered independently) represents such an increase as the reanalyses currently do.

Finally, to better illustrate the situation, we would like to add 2D graphs of situations representing 1) Greenland over the last 20 years affected by recurrent summer blocking events, 2) the non representation of this type of situation by the CMIP6, and the non projection of such situations.

To do this, we removed the mean Z500 of the area considered (northern hemisphere, 90-50° N) in order to isolate the « basic » Z500 from the global increase in Z500 due to warming (called ZG2 hereafter). Indeed, if we consider "raw" anomalies of future Z500 with respect to a reference period, we end up with stifled information in the increase in Z500 caused by the respective warming of each ESM since air temperature increase will lead to an increase in geopential height (e.g. Fig 2 for EC-Earth3). By removing the average over the considered area, we mainly remove the signal coming from temperature increase by assuming here that the warming is almost uniform over the considered area. Finally, we compared this average ZG2 over the critical period of the NCEP (2000-2019) with that of the reference period (1970-1999), which was not characterised by summer blocking events (fig 3a). We also applied this analysis to the ERA5, which show a strong similarity to the NCEP reanlaysis (Fig 3b). The same argument was also applied to two CMIP6 ESMs. Fig 4a (5a respectively) therefore compares the NESM3 ZG2 anomalies (resp. MRI-ESM2-0) for the period 2000-2019 with respect to the reference period, and Fig 4c (5b respectively) compares the reference period ZG2 to 2080-2100. These ESMs were already highlighted in the manuscript as they have higher values of GB2.

For NESM3, the GB2 anomaly for the period 2030-2050 was also compared to the reference period (Fig 4b) given the values of normalized GB2 very close to unity in Figure 2 of the paper. The 2D comparison does show a positive ZG2 anomaly over the Baffin Island but not significant relative to the variability of ZG2 during the reference period, which is not the case for ZG2 anomalies derived from reanalysis over Greenland. It is exactly the same case for MRI-ESM2-0 over 2000-2019 over Greenland (Fig 5a), but the anomalies are also not sgnificant.

All of these figures illustrate the fact that the reanalyses represent an increase in blocking situations inferred from an increase in the Z500 located in Greenland, whereas the ESMs show neither a significant increase for the current period nor for the future.

[Figure]

Fig 2. Z500 anomalies (m) over 2080-2100 from EC-Earth3 with respect to reference period (1970-1999).

[Figure]

Fig 3a. ZG2 anomalies (m) over 2000-2019 for NCEP reanalysis with respect to the reference period (1970-1999). ZG2 was computed by removing the mean Z500 between 50°N and 90°N to remove the artificial increase in geopotential height due to global warming.

[Figure]

Fig 3b. Similar ton Fig 3a but for ZG2 (m) computed with ERA5 reanalyses.

[Figure]

Fig 4a. Similar ton Fig 3a but for ZG2 (m) computed with NESM3 ESM.

[Figure]

Fig 4b. Similar to Fig 3a but for ZG2 (m) computed with NESM3 ESM over 2030-2050. The period 2030-2050 is displayed as NESM3 has a "high" GBI index during this period but the pattern remains different compared to the observed current blocking pattern over Greenland.

[Figure]

Fig 4c. Similar to Fig 3a but for ZG2 (m) computed with NESM3 ESM over 2080-2100. The figure points out that there is no blocking increase in NESM3 projections for the end of the century.

[Figure]

Fig 5a. Similar to Fig 3a but for ZG2 (m) computed with MRI-ESM2-0.

[Figure]

Fig 5b. Similar to Fig 3a but for ZG2 (m) computed with MRI-ESM3-0 ESM over 2080-2100.

We understand the reviewers' comments and yours, but we hope that our answers will make you reconsider your decision and allow us to revise our manuscript. In addition to adding more reanalyses, 2D illustrations of blockings events, we obviously agree to change the title of the paper to something that only discusses anticyclonic blockings such as "CMIP6 does not suggest any increase of blocking event in summer over Greenland by 2100".

Reference : Hanna, E., Fettweis, X., and Hall, R. J.: Recent changes in summer Greenland blocking captured by none of the CMIP5 models, The Cryosphere, pp. 3287–3292, https://doi.org/10.5194/tc-12-3287-2018, 2018.